# Patient-reported factors associated with avoidance of in-person care during the COVID-19 pandemic: Results from a national survey

**Bengt B. Arnetz**[1][☯]*, **Courtney Goetz**[1][☯][¤], **John vanSchagen**[1,2][‡], **William Baer**[2][‡], **Stacy Smith**[2][‡], **Judith E. Arnetz**[1][☯]

**1** Department of Family Medicine, College of Human Medicine, Michigan State University, Grand Rapids, Michigan, United States of America, **2** Trinity Health Saint Mary's, Grand Rapids, Michigan, United States of America

☯ These authors contributed equally to this work.
¤ Current address: Department of Psychology, College of Humanities and Social Sciences, Louisiana State University, Baton Rouge, Louisiana, United States of America
‡ These authors also contributed equally to this work
* arnetzbe@msu.edu

**Data Availability Statement:** The relevant data is held in Figshare at https://doi.org/10.6084/m9. figshare.19438469.v1 (https://figshare.com/

## Abstract

### Background

There has been a substantial decline in in-person care in inpatient and outpatient settings during the ongoing COVID-19 pandemic. Avoidance of needed in-person care may contribute to an avoidable decline in patient health and an increase in mortality. While several systems and behavioral theories have been put forward to explain the decline, there is a lack of studies informed by patients' own experiences. The current study applied a socio-ecological model encompassing patient, environmental, and institutional-related variables to examine patient-reported factors associated with avoidance of in-person care.

### Methods

Between October and December 2020, a total of 3840 persons responded to a nationwide online questionnaire that was administered using ResearchMatch and Facebook. Self-reported avoidance of in-person care among those who needed it was the main outcome. Multivariable logistic regression analysis was used to identify factors associated with avoidance of needed care.

### Findings

Out of a total of 3372 respondents who reported that they needed in-person care during the early phase of the pandemic, 257 (7.6%) avoided it. Patient-related variables associated with avoiding needed care included younger age (odds ratio (OR), 1.46, 95% CI 1.11 to 1.94, p<0.01; <45 y/o vs 45+), inability to afford care (OR = 1.65, 95% CI 1.17 to 2.34, p<0.01), and greater COVID-related stress (OR = 1.36, CI 1.01 to 1.83, p<0.05). More

articles/journal_contribution/Covid_19_Care_
avoidance_Arnetz/19438469).

**Funding:** BA, JA and CG was funded by a grant
from Mercy Saint Mary's. NO - The funders had no
role in study design, data collection and analysis,
decision to publish, or preparation of the
manuscript.

**Competing interests:** The authors have declared
that no competing interests exist.

frequent discussions about COVID with family and friends was the only significant environment-related avoidance of care variable (OR = 1.39, 95% CI 1.01–1.91, p < .05). Institution-related care avoidance variables included a negative patient healthcare experience rating (OR 1.83, 95% CI 1.38 to 2.42, p<0.001), poor awareness of the institution's safety protocol (OR = 1.79, 95% CI 1.28 to 2.51, p<0.01), and low ratings of the institution's effectiveness in communicating their safety protocol (OR = 3.45, 95% CI 1.94 to 6.12, p<0.001). The final model predicted 11.9% of the variance in care avoidance.

## Conclusions

These results suggest that care avoidance of in-person care during the initial phase of the pandemic was influenced by a patient's demographics as well as environmental and healthcare institutional factors. Patients' previous experiences and their awareness of healthcare systems' safety protocols are important factors in care avoidance.

## Introduction

There was a well-documented precipitous decline in both in- and out-patient in-person healthcare visits during the initial phase of the COVID-19 pandemic [1–3]. While some of these visits might have been converted to telemedicine, the decline in in-person care may have contributed to increased rates of avoidable patient deaths and delayed attention to existing acute and chronic diseases [4–7]. Studies of the current pandemic and prior wide-spread infectious outbreaks have informed our understanding of the role of demographic and attitudinal factors in a person's decision to adopt preventive (e.g., wearing masks, social distancing, or taking vaccines), avoidance (avoiding mass transit and hospital visits), and/or disease-managing behaviors (taking antivirals, consulting healthcare professionals) [8–11]. Recent research identified personality factors such as risky decision-making, risk perception, and temporal discounting (i.e., opting for smaller immediate rewards instead of larger delayed ones) as predictive of compliance with the COVID-19 preventative behaviors of mask-wearing and social distancing [11]. The current paper focuses specifically on avoidance behavior related to in-person medical care.

The literature on care avoidance during the COVID-19 pandemic has primarily been based on retrospective analyses of hospital admissions [1] or emergency care visits [3]. A study of patterns in hospital admissions found declines of more than 20% for all diagnoses from February to April 2020 [1]. In the first 10 weeks of the pandemic, emergency department (ED) visits for heart attack decreased by 23%, stroke by 20%, and hypoglycemic crisis by 10% [12]. This was corroborated by a retrospective study of 162 ED sites that reported declines in visits for myocardial infarction (AMI), stroke and sepsis, especially among older patients, between January 2019 and November 2020 [3]. A June 2020 article showed that there was a 26% decrease in emergency medical services (EMS) activations in the early months of 2020, even as EMS-attended deaths doubled [13]. In a study of Medicare beneficiaries, hospitalization for myocardial infarction and stroke decreased 14% in 2020 compared to the prior two years [14]. Among Medicare beneficiaries with 6 or more chronic conditions, the decline was 42%. By the end of June 2020, the CDC estimated that 41% of US adults avoided emergency or routine care [7]. At the same time, patient use of telehealth services increased dramatically. For example, one large institution saw telehealth visits increase from less than 1% of total visits to 70% of total

visits during a four-week period in March-April 2020 [15]. In a study in Detroit, MI, patients living in census tracts with higher vulnerability exposures, e.g., greater proportion of population living in poverty, being unemployed, or not proficient in the English language, were more likely to voluntarily refuse EMS transport to hospital emergency care against medical advice during the pandemic as compared to before [16]. Moreover, during the peak of the pandemic, the risks for prehospital deaths, excluding in-transit deaths, were significantly higher in these socioeconomically stressed census tracts [16].

Several studies during the first year of the pandemic highlighted the impact of COVID-19 on patient care and health outcomes for specific diseases, including myocardial infarction and stroke [14], various pediatric conditions [17], hypertension [18], Alzheimer's [19], Parkinson's [20], and other neurological diseases [21]. All these studies mention the lack of access to providers and hospitals due to diversion of resources focused on COVID-19 care; these are structural factors, not factors of patient behavior. The studies focused on neurological conditions were mainly concerned with the vulnerability of patients to COVID-19 infection [19] and the potential aftermath of infection on cognitive ability and neurological function [20,21]. However, research on myocardial infarction, stroke [14], and emergency pediatric conditions [17] pinpointed patient/caregiver avoidance of needed care as a possible risk factor for an increase in adverse outcomes, although cause and effect could not be determined [14,17]. A recent study provides an overview of the impact of the pandemic on oncology care, citing ten main impact categories ranging from reduced access to medical equipment and medication to patients' psychological health. Patients' fear of COVID-19 infection and lockdowns are cited as two main reasons for patient avoidance of scheduled cancer screenings [22].

However, there is only limited research into the reasons for care avoidance of needed in-person care during COVID-19. In an April 2020 Gallup poll, 83% of respondents reported that they would be at least 'moderately concerned' about being exposed to COVID-19 at a doctor's office or hospital. [23] In May 2020, a study of pediatric hospitals in Italy was one of the first to suggest that fear of contagion was a predictor of care avoidance [17]. A study from June 2020 from the COVID-19 Outbreak Public Evaluation (COPE) Initiative reported that 40.9% of respondents to an online survey had avoided medical care. However, respondents' reasons for avoiding care were not reported [7]. Another study of U.S. adults from June 2020 was based on the Household Pulse Survey, administered by the U.S. Census Bureau and other federal agencies to continuously document the impact of the COVID-19 pandemic on U.S. households. The survey asked adults whether they had delayed and/or avoided seeking medical care, even if it were needed, during the COVID-19 pandemic [24]. Results showed that 41% delayed seeking medical care, one third avoided it completely, and symptoms of anxiety and depression were strongly correlated with medical care avoidance [24].

Care avoidance has been studied during previous outbreaks, such as the H1N1 pandemic [8,10,25]. These studies pointed to individual factors such as age, gender [25], marital status and having small children [10], as well as beliefs, lack of accurate information [25], personality traits and mistrust of authorities [8,10] that influenced avoidant behaviors. These studies examined care avoidance activities related to in-person care (e.g., public transportation) but did not focus specifically on avoidance of needed medical care. Few, if any, focused on reasons provided by patients for avoiding needed in-person care. While patient-level factors are important, patients' beliefs and behavior are closely integrated within a more complex socioecological framework. Based on this literature review, we hypothesized that factors related to the individual patient, the patient's environment, and their healthcare institution would all play a role in patients' avoidance of care regarding healthcare visits. This study used Bronfenbrenner's socio-ecological model, which is endorsed by the Centers for Disease Control, as a theoretical framework (Fig 1) [26,27]. Building on the CDC's four-level model of violence

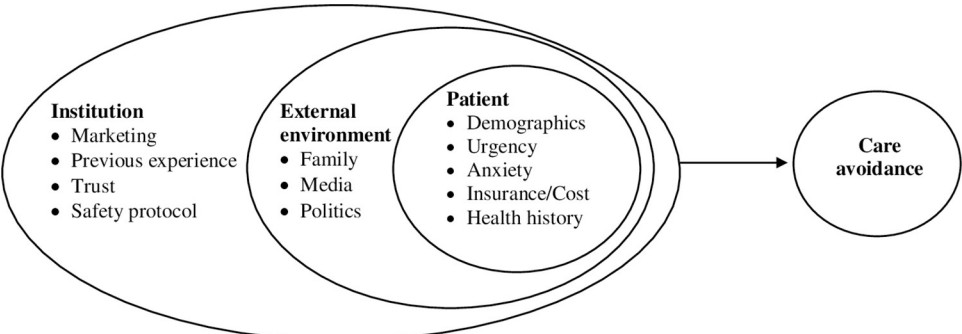

**Fig 1. Conceptual model of care avoidance.**

prevention [27], the model in the current study includes three levels representing patient, environment, and institutional factors, respectively, as potential predictors of in-person care avoidance in inpatient and outpatient settings. Based on the socioecological model, the individual is the innermost of three concentric circles, with demographic and other individual factors representing the closest sphere influencing care seeking/avoidance behaviors. This is followed by the individual's most immediate environment (family, community), represented by the second circle. Finally, the outermost circle in our proposed model of care avoidance is the healthcare institution where the individual would normally seek care [28].

We hypothesized that patient variables associated with care avoidance included demographics, age, gender, and anxiety related to the COVID-19 pandemic [9,10,17,18,24,29]. Factors related to the patient's immediate environment were hypothesized to include the viewpoints expressed by family members, the media, and political discourse. Finally, we hypothesized that factors related to the patient's healthcare organization, such as the patient's previous experiences with the entity, trust in the institution, and possible media campaigns, might be related to avoidance of care [16,18].

## Methods

### Study design

A cross-sectional online survey was conducted in the United States from late October to early December 2020.

### Participants

Participants were recruited nationwide through Facebook and ResearchMatch [30]. Those who expressed interest were sent a Qualtrics link containing the survey. The sole inclusion criterion was being 18 or older, and there were no exclusionary criteria. Participants completed consent as the first page of the survey and were informed that the survey was anonymous. The study was determined exempt by the Michigan State University Institutional Review Board.

### Measures

The study instrument consisted of a 45-item online survey that was administered via Qualtrics. The following variables were used in the current study:

**Outcome variable.** <u>Care avoidance</u> was assessed by the question: Have you had an in-person visit with any care provider during the pandemic?' Responses were 'Yes, I have had an in-person visit', 'No, I avoided this type of care', and 'No, I didn't need this type of care'. For this

study, the outcome of interest was avoidance of in-person care among those who needed it; analyses therefore did not include telehealth visits.

**Independent variables.**  Based on our socioecological conceptual model, independent variables were categorized as either patient, environment, or institutional variables.

**Patient variables.**  *Demographics*. Basic demographic information was collected including age, gender identity, race/ethnicity, and employment status.

*Health variables*. Health-related variables included factors related to healthcare access as well as individual health. Variables related to healthcare access included health insurance status, 'Do you currently have health insurance?' (yes/no); ability to afford healthcare services, 'I can currently afford to pay for healthcare services.' (true/false); and access to transportation to healthcare providers, 'If needed, I could access transportation to a healthcare facility (driving your own car, using public transportation, etc.)' (true/false). Individual health variables included chronic illness diagnosis, 'Have you been diagnosed with any chronic health conditions (for example–diabetes, high blood pressure, cancer, arthritis, etc.)?' (yes/no). General health and mental or emotional health were assessed by asking, 'In general, how would you rate your overall health (or mental or emotional)? Responses to both the latter two health ratings were on a five-point scale from 1'Poor' to 5 'Excellent,' with higher scores indicating better health. Participants were also asked about COVID-19 diagnosis ('Have you been diagnosed with COVID-19?' yes/no) and/or symptoms ('Have you had symptoms that make it highly likely that you have had COVID-19? Yes/no).

*COVID-related stress*. To assess fear of contagion and other stressors attributable to the COVID pandemic, we used the 'danger' scale (5 items) and a selection of 4 items from the 'contamination' scale (questions related to banking/cash transactions were excluded to keep the survey brief) from the validated COVID Stress Scales [29]. Patients rated these nine statements about their COVID-related fears, e.g. 'I am worried that basic hygiene (e.g., hand washing) is not enough to protect me from the virus,' on a five-point scale from 'Not at all', scored 1 to 'Extremely', scored 5. Higher scores indicate higher levels of COVID-related stress. Responses to each question were summed and the total score was used to indicate levels of COVID-related stress. Cronbach's alpha for the overall scale was 0.92.

**Environment variables.**  Variables related to patients' immediate environment included reports of how often they discussed COVID-19 with family and friends, consumed COVID-related news and social media, and read about or discussed COVID-related politics using a five-point Likert scale from "Never" (1) to "Very often" (5).

**Institutional variables.**  *Marketing*. Many healthcare organizations have been using marketing to inform their patients that it is safe to return to in-person care, sometimes by explaining the safety protocols being using in their hospitals/clinics. Patients reported whether they had seen advertising from their healthcare organization about returning to in-person care, and how often they had seen advertisements using a four-point Likert scale from "Never" to "Many times". Those who had seen ads reported the type of advertisements (television, signage, billboard, etc.).

*Previous experience*. The patients' previous experience with their primary healthcare organization was measured using two questions adapted from the Hospital Consumer Assessment of Healthcare Providers and Systems (HCAHPS) survey: a general experience rating ('Overall, how would you rate your experience with the health care organization you use most often?') on a sliding scale from very negative to very positive, and willingness to recommend ('I would be willing to recommend the health care organization I use most often to my family and friends') on a sliding scale from completely disagree to completely agree [31].

*Safety protocol/Institutional trust*. The survey included three questions about the safety protocols being used in hospitals and clinics to prevent the spread of COVID-19. Patients reported

whether they were aware of these protocols, whether the protocol made them feel safer (representing effective safety protocol communication), and whether they believed the protocol was in fact being followed (representing trust in the healthcare organization). Questions were answered on a three-point scale from "Not at all" to "Completely".

*Feelings of safety*. Patients rated their feelings of safety for primary care, inpatient care, and outpatient care visits, respectively, using the following format: 'Do you feel safe visiting your primary care provider in the midst of the pandemic? (sliding 0–10 scale from 'not at all' to 'completely', translated to a percentage for analysis)'.

### Analysis

Statistical analysis was conducted using IBM SPSS Statistics version 26 (IBM Corp, Armonk, NY). A two-sided P value <0.05 represented statistical significance. A principal component analysis of the three safety rating variables gave a single factor solution with an alpha of .88, thus justifying the combination of the variables into an index. Hence, a combined Total Safety Rating Scale was used as a predictor variable. All predictor variables were transformed into dichotomous categorical variables using either established cutoff scores, means (for continuous variables), or medians (for categorical variables). Predictor variables that were not independently related to the outcome variable in a Chi-square analysis were excluded from further analyses.

Multivariable logistic regression analysis was performed to identify factors associated with avoidance of in-person care among those needing it. In keeping with our socioecological model, patient variables (gender identity, age, race/ethnicity, ability to afford care, COVID-19 symptoms, COVID-related stress) were added in the first step, environment variables (discussing COVID with family/friends) were added in the second step, and institutional variables (return-to-care advertising exposure, prior experience of the healthcare organization, awareness of organization's safety protocol, safety protocol communication effectiveness, believing safety protocol is followed) were added in the third step. Collinearity diagnostics were performed, and variables were removed accordingly until acceptable variance inflation factors (<2), Eigenvalues (approaching 1), and condition index values (<30) were reached.

### Results

A total of 3840 patients completed the survey. Patients who reported that they had not needed care (n = 423, 11.0%) and those not responding to the question (n = 45, 1.2%) were excluded from this analysis, leaving a sample size of 3373. Participant characteristics are summarized in Table 1.

For all demographic variables, missing values ranged from 0–0.36%. Patients ranged from 18 to 75+ years of age, with the median age group 55–64 years. Most respondents were female (74.6%), White (87.2%), and employed (52.0%). Among those who were furloughed or unemployed, 4.1% (n = 137) reported that it was due to the pandemic. While most patients could afford health care costs (88.9%), 11.1% of patients reported they could not currently afford to pay for health care. Most patients reported that they had an in-person visit during the pandemic (92.4%), while 7.6% (n = 257) reported that they had avoided care.

Patients who avoided care differed significantly from those who sought care on patient, environmental, and institution-related variables (Table 2).

Ten of the 15 patient-related variables were significantly associated with care avoidance. Patients who were younger ($\chi^2$ = 35.54, p < .001), did not identify as male or female ($\chi^2$ = 8.09, p<0.05), and non-White ($\chi^2$ = 3.96, p < .05) were more likely to avoid care. Those who were retired were less likely to avoid care ($\chi^2$ = 31.13, p<0.001), while those who were unemployed

**Table 1. Demographics of study participants (n = 3372).**

| | | N*(%†) |
|---|---|---|
| **Gender identity** | Female | 2514 (74.6) |
| | Male | 771 (22.9) |
| | Additional identity | 75 (2.2) |
| **Age** | 18–24 | 127 (3.7) |
| | 25–34 | 480 (14.2) |
| | 35–44 | 449 (13.3) |
| | 45–54 | 466 (13.8) |
| | 55–64 | 733 (21.7) |
| | 65–74 | 830 (24.6) |
| | 75+ | 286 (8.5) |
| **Race/ethnicity** | White | 2942(87.2) |
| | Non-white | 430 (12.8) |
| **Employment status** | Employed | 1752 (52.0) |
| | Furloughed | 40 (1.2) |
| | Unemployed | 349 (10.3) |
| | Retired | 1231 (36.5) |
| **Ability to afford care** | Can afford health care costs | 2996 (88.8) |
| | Cannot afford health care costs | 373 (11.1) |
| **In-person care** | Had in person visit | 3115 (92.4) |
| | Avoided in-person care | 257 (7.6) |

*Numbers do not add to total sample size due to missing values.

†Valid percentages are reported.

were more likely to avoid care. Those who lacked health insurance ($\chi^2$ = 25.33, p < .001), could not afford healthcare ($\chi^2$ = 22.04, p<0.001), and/or lacked access to transportation ($\chi^2$ = 7.88, p<0.01) were also more likely to avoid care. Patients who experienced COVID-19 symptoms were more likely to avoid care ($\chi^2$ = 7.93, p<0.01), as were those who reported poorer mental health ($\chi^2$ = 19.07, p<0.001) and/or higher COVID-related stress ($\chi^2$ = 35.69, p<0.001).

Environmental variables associated with care avoidance were more frequent discussions of COVID-19 with family and friends ($\chi^2$ = 9.63, p < .01) and more frequent exposure to COVID-related politics ($\chi^2$ = 5.27, p < .05).

All seven institutional variables were associated with care avoidance. Lower exposure to return-to-care advertising ($\chi^2$ = 8.09, p < .01) and lower ratings on: the respondents' experience with their healthcare organization ($\chi^2$ = 58.40, p<0.001), willingness to recommend ($\chi^2$ = 58.06, p < .001), total safety ($\chi^2$ = 134.52, p<0.001), awareness of safety protocol ($\chi^2$ = 47.43, p<0.001), safety protocol communication effectiveness ($\chi^2$ = 57.75, p<0.001), and belief that safety protocol is followed ($\chi^2$ = 33.94, p<0.001) were all related to care avoidance.

The following variables were excluded from the regression model due to lack of independent relationship with care avoidance: needed urgent care, needed emergency care, chronic conditions, COVID-19 diagnosis, general health rating, consuming COVID-related content on social media, and consuming COVID-related content on news media. The final logistic regression analysis for factors associated with care avoidance is shown in Table 3.

In instances of multicollinearity, the variable with the weaker relationship to the outcome variable was excluded from the model. Variables excluded from the regression model due to multicollinearity were employment status, health insurance status, transportation access,

**Table 2. Chi-square examining independent variables associated with care avoidance (n = 3372).**

| | Avoided care | Did not avoid care | $\chi^2$ |
|---|---:|---:|---|
| | N (%) | N (%) | |
| *Patient-related variables* | | | |
| Gender identity | | | |
| Male | 53 (20.7) | 718 (23.1) | 8.09* |
| Female | 191 (74.6) | 2323 (74.8) | |
| Additional identity | 12 (4.7) | 63 (2.0) | |
| Age | | | |
| Younger than 45 | 134 (52.1) | 932 (29.9) | 35.54*** |
| 45+ | 123 (47.9) | 2183 (70.1) | |
| Race/ethnicity | | | |
| White | 214 (83.3) | 2728 (87.6) | 3.96* |
| Non-White | 42 (16.7) | 387 (12.4) | |
| Employment status | | | |
| Employed/Self-employed | 146 (56.8) | 1606 (51.6) | 31.13*** |
| Furloughed | 5 (1.9) | 35 (1.1) | |
| Unemployed | 46 (17.9) | 303 (9.7) | |
| Retired | 60 (23.3) | 1171 (37.6) | |
| Health insurance status | | | |
| Uninsured | 17 (6.6) | 57 (1.8) | 25.33*** |
| Insured | 240 (93.4) | 3068 (98.2) | |
| Ability to afford care | | | |
| Cannot afford | 51 (19.9) | 322 (10.3) | 22.04*** |
| Can afford | 205 (80.1) | 2791 (89.7) | |
| Transportation access | | | |
| Cannot access | 8 (3.1) | 34 (1.1) | 7.88** |
| Can access | 249 (96.9) | 3080 (98.9) | |
| Needed emergency care | | | |
| No | 201 (78.5) | 2550 (82.0) | 1.92 |
| Yes | 55 (21.5) | 560 (18.0) | |
| Needed urgent care | | | |
| No | 165 (64.2) | 1925 (61.8) | 0.57 |
| Yes | 92 (35.8) | 1189 (38.2) | |
| Diagnosed w/ chronic condition(s) | | | |
| No | 103 (40.1) | 1064 (34.2) | 3.68 |
| Yes | 154 (59.9) | 2051 (65.3) | |
| Diagnosed w/ COVID-19 | | | |
| No | 253 (98.4) | 3033 (97.4) | 1.11 |
| Yes | 4 (1.6) | 82 (2.6) | |
| COVID-19 symptoms | | | |
| No | 218 (84.8) | 2812 (90.3) | 7.93** |
| Yes | 39 (15.2) | 301 (9.7) | |
| General health rating (Range 1–5, median = 3) | | | |
| $\leq 3$ | 142 (55.3) | 1582 (50.8) | 1.90 |
| $> 3$ | 115 (44.7) | 1533 (49.2) | |
| Mental health rating (Range 1–5, median = 3) | | | |

*(Continued)*

**Table 2.** (*Continued*)

| | Avoided care | Did not avoid care | $\chi^2$ |
|---|---|---|---|
| | N (%) | N (%) | |
| ≤3 | 177 (68.9) | 1707 (54.8) | 19.07*** |
| >3 | 80 (31.1) | 1408 (45.2) | |
| COVID-related stress (Range 0–36, mean = 15) | | | |
| ≤15 | 85 (33.1) | 1634 (52.5) | 35.69*** |
| >15 | 172 (66.9) | 1481 (47.5) | |
| *Environment-related variables* | | | |
| Discussing COVID w/ family and friends (Range 1–5, median = 3) | | | |
| ≤3 | 66 (25.9) | 1099 (35.5) | 9.63** |
| >3 | 189 (74.1) | 1996 (64.5) | |
| Consuming COVID-related social media content (Range 1–5, median = 2) | | | |
| ≤2 | 139 (54.5) | 1863 (60.2) | 3.17 |
| >2 | 116 (45.5) | 1232 (39.8) | |
| Consuming COVID-related news media (Range 1–5, median = 3) | | | |
| ≤3 | 103 (40.4) | 1296 (41.9) | 0.22 |
| >3 | 152 (59.6) | 1796 (58.1) | |
| Exposure to COVID-related politics (Range 1–5, median = 3) | | | |
| ≤3 | 174 (68.2) | 2312 (74.8) | 5.27* |
| >3 | 81 (31.8) | 780 (25.2) | |
| *Institution-related variables* | | | |
| Return-to-care advertising exposure (Range 0–6, median = 2) | | | |
| ≤2 | 182 (71.1) | 1925 (62.2) | 8.09** |
| >2 | 74 (28.9) | 1172 (37.8) | |
| HCO experience rating (Range 0–100, mean = 81) | | | |
| <81 | 159 (61.9) | 1172 (37.6) | 58.40*** |
| ≥81 | 98 (38.1) | 1943 (62.4) | |
| Willingness to recommend HCO (Range 0–100, mean = 81) | | | |
| <81 | 150 (58.4) | 1077 (34.6) | 58.06*** |
| ≥81 | 107 (41.6) | 2038 (65.4) | |
| Total safety rating (Range 0–300, mean = 173) | | | |
| ≤173 | 205 (83.7) | 1377 (45.2) | 134.52*** |
| >173 | 40 (16.3) | 1670 (54.8) | |
| Awareness of HCO safety protocol | | | |
| Unaware | 60 (23.5) | 299 (9.7) | 47.43*** |
| Aware | 195 (76.5) | 2798 (90.3) | |
| Safety protocol communication effectiveness | | | |
| Do not feel completely safe | 243 (94.6) | 2278 (73.1) | 57.75*** |
| Feel completely safe | 14 (5.4) | 837 (26.9) | |
| Believe safety protocol is followed | | | |

(*Continued*)

**Table 2.** (Continued)

| | Avoided care | Did not avoid care | $\chi^2$ |
|---|---|---|---|
| | N (%) | N (%) | |
| Do not completely believe | 178 (69.3) | 1569 (50.4) | 33.94*** |
| Completely believe | 79 (30.7) | 1546 (49.6) | |

*p < .05

**p < .01

***p < .001.

mental health rating, exposure to COVID-related politics, willingness to recommend, and total safety rating. Patient-related variables associated with an increased likelihood of care avoidance included younger age (OR = 1.78, 95% CI 1.35 to 2.34, p<0.001), inability to afford care (OR = 1.91, 95% CI 1.36 to 2.68, p<0.001), and higher COVID-related stress (OR = 1.93, CI 1.46 to 2.54, p<0.001). These variables explained 5.4% of the variance as estimated using Nagelkerke $R^2$. In the next step, the environment variable, discussing COVID-19 with family and friends, was not significantly associated with care avoidance but the three patient-related variables remained significant. The explained variance was 5.6% following step two. In step three, the institutional variables–lower healthcare organization experience ratings (OR 1.83, 95% CI 1.38 to 2.42, p<0.001), poor awareness of the organization's safety protocol (OR = 1.79, 95% CI 1.28 to 2.51, p<0.01), and lower safety protocol communication effectiveness (OR = 3.45, 95% CI 1.94 to 6.12, p<0.001)–were associated with an increased likelihood of care avoidance. The environment variable, discussing COVID-19 with family and friends,

**Table 3. Logistic regression for factors associated with care avoidance (n = 3372).**

| | Step 1 | | Step 2 | | Step 3 | |
|---|---|---|---|---|---|---|
| | OR | 95% CI | OR | 95% CI | OR | 95% CI |
| Gender identity [Ref. Male] | | | | | | |
| Female | 0.87 | 0.63 to 1.21 | 0.85 | 0.61 to 1.18 | 0.89 | 0.64 to 1.25 |
| Additional identity | 1.15 | 0.55 to 2.39 | 1.16 | 0.56 to 2.42 | 1.16 | 0.55 to 2.45 |
| **Age younger than 45** [Ref. 45 and older] | **1.78***** | **1.35 to 2.34** | **1.76***** | **1.34 to 2.32** | **1.46**** | **1.11 to 1.94** |
| Race/ethnicity [Ref. white] | 1.25 | 0.87 to 1.78 | 1.27 | 0.89 to 1.81 | 1.33 | 0.93 to 1.91 |
| **Ability to afford care** [Ref. yes] | **1.91***** | **1.36 to 2.68** | **1.96***** | **1.39 to 2.75** | **1.65**** | **1.17 to 2.34** |
| COVID-19 symptoms [Ref. no] | 1.37 | 0.94 to 1.99 | 1.35 | 0.93 to 1.97 | 1.30 | 0.88 to 1.90 |
| **COVID-related stress** [range 0–36, mean = 15, Ref. ≤15] | **1.93***** | **1.46 to 2.54** | **1.78***** | **1.34 to 2.38** | **1.36*** | **1.01 to 1.83** |
| **Discussing COVID w/ family and friends** [range 1–5, median = 3, Ref. ≤3] | | | 1.33 | 0.97 to 1.81 | **1.39*** | **1.01 to 1.91** |
| Return-to-care advertising exposure [range 0–6, median = 2, Ref. ≤ 2] | | | | | 0.75 | 0.56 to 1.01 |
| **HCO**[†] **experience rating** [range 0–100, mean = 81, Ref. ≥ 81] | | | | | **1.83***** | **1.38 to 2.42** |
| **Awareness of HCO safety protocol** [Ref. yes] | | | | | **1.79**** | **1.28 to 2.51** |
| **Safety protocol communication effectiveness** [Ref. completely safe] | | | | | **3.45***** | **1.94 to 6.12** |
| Believing safety protocol is followed [Ref. completely believe] | | | | | 1.18 | 0.87 to 1.60 |
| Nagelkerke $R^2$ | **0.054***** | | 0.056 | | **0.119***** | |

*p≤.05

**p≤.01

***p≤.001.

[†]Health care organization.

also became significant in this step (OR = 1.39, 95% CI 1.01 to 1.01, p<0.05). Return-to-care advertising exposure and believing safety protocol is followed were not associated with care avoidance. This final model had an explained variance of 11.9%.

## Discussion

This study focused on understanding why patients avoided needed in-person care during the early phase of the COVID-19 pandemic. This was an exploratory study, and many of the variables we initially hypothesized to have a relationship with care avoidance, such as prevalence of chronic conditions and increased media exposure, were not related to care avoidance. However, there was considerable multicollinearity among those variables that were related to avoiding in-person care. Thus, we were eventually driven towards a more parsimonious model with less noise.

Nearly 8% of our sample (n = 257) reported that they had avoided necessary in-person care. This is a much smaller proportion compared to two previous questionnaire-based studies, one of which reported 41% of respondents had avoided care [7] and the other reporting 32% [24]. Both of those studies were based on nationally representative samples, were conducted during a one-week period in June of 2020 and asked about avoidance of any type of medical care. In contrast, the current study utilized a convenience sample, was conducted over a 3-month period, October to-December 2020, and asked about avoidance of in-person care that patients deemed necessary. Czeisler and colleagues [7] combined delayed or avoided care into a single variable, asking only whether delay/avoidance was due to concerns about COVID-19. Ganson and colleagues [24] asked about delay and avoidance in the past 4 weeks. These methodological differences make comparisons between studies difficult and may explain the lower percentage of care avoidance in our study, since we asked specifically about avoidance of needed in-person care only. Ganson and colleagues [24] focused on the role of anxiety and depression in care avoidance. Neither of the other studies examined the influence of media, family members, or the patient's prior healthcare experience, all variables in our study.

Care avoidance in the logistic regression was associated with higher COVID-related stress and demographic variables such as younger age and inability to afford care, but also institutional variables–lower ratings of previous experience with their health care organization and poor communication of safety protocols. The pandemic may in fact have exacerbated the inability to afford care and care avoidance for some respondents, as approximately 4% of those who were furloughed or unemployed reported that it was due to the pandemic. While patient-related variables explained about 5.4% of the variance in care avoidance, institutional variables explained more than 6%. More frequent discussions of COVID-19 with family and friends were associated with an increased likelihood of care avoidance. In contrast, no other variables related to the patients' external environment—media consumption and discussion of COVID-related politics–predicted avoidance of care.

Our model included demographic factors, specifically those traditionally linked to healthcare inequity such as race/ethnicity and gender, as potential influences on care avoidance. Although younger age was significant in our analyses, this effect was small. This finding is in contrast to prior work on avoidance behavior during pandemics, where typically older persons refrained from care [8]. However, care avoidance was significantly higher among younger and non-White individuals both in our study and in a study based on a U.S. nationally representative sample [7]. One of the impactful patient-related variables in each of our models was COVID-related stress. Patients were more likely to avoid care if they scored higher on this scale, which measures fear of contagion and concern about protecting loved ones from the virus. This echoes one Italian study [17] and one nationally representative U.S. study [24]

which related fear and anxiety with care avoidance. This effect was also shown in previous pandemics such as SARS-CoV-1 and H1N1 [8,10,25].

Our hypothesized model also included environment-related variables such as consuming COVID-related media news and the politics related to the pandemic, as we reasoned they might instill fear and panic, thereby influencing care avoidance. In fact, these factors had no significant impact on the avoidance of care and were outweighed by institutional and demographic factors. The only environment-related variable that was significant in our final regression model was discussing COVID-19 with family and friends. Patients who reported discussing COVID with family and friends more often were more likely to avoid care. Nevertheless, the variables concerning COVID-related news media and politics were significantly correlated with COVID-related stress scores and may have contributed to COVID-related stress and thus indirectly to care avoidance.

During the Middle East Respiratory Syndrome (MERS) pandemic, health-related communications were shown to mediate the relationship between social determinants of health and care avoidance [32]. While advertising exposure did not have a significant direct impact on patients' care seeking, results suggest that effective communication of the healthcare organization's safety protocol could help to improve the efficacy of advertisements, as patients were less likely to avoid care if they knew of their healthcare organization's safety protocol, and even less likely when knowledge of the safety protocol made them feel safe. In fact, patients who reported that the safety protocol did not make them feel completely safe were about 3.5 times more likely to avoid care. Findings suggest that healthcare organizations need to design media messages and advertisements that are more specifically related to safety protocols, and more general messages are unlikely to make patients feel confident seeking in-person care.

Another important finding is that patients' already established experiences with their healthcare organization significantly influenced care avoidance behavior. Patients who reported having had a negative overall experience with their healthcare organization were 1.83 times more likely to avoid care. This suggests that striving to enhance patients' experience and trust in their healthcare organization in general is important to ensure that patients seek needed care during public health crises such as pandemics. It is well established that patient experiences are associated with objective measures of healthcare outcomes [33]. As discussed earlier, care avoidance is also related to adverse health outcomes in patients with established acute and chronic diseases, including increased mortality [7,12].

Finally, it is important to note that the strain of providing care for COVID-patients varied for healthcare organizations, both hospitals and outpatient clinics, at different points during the pandemic [34,35]. In a February 2021 pulse survey among 320 U.S. hospitals, the Office of Inspector General's office reported that hospital resources had been severely strained by the burden of caring for COVID-19 patients, and that caring for patients who had delayed or avoided care would be a major challenge going forward [35]. By the end of 2020, The Commonwealth Fund reported that outpatient visits had begun to stabilize after a significant dip early on in the pandemic [36]. However, our survey did not ask participants whether their avoidance of in-person care was due to resource diversion for prioritization of COVID patients at their healthcare organizations. Instead, we focused on participant exposure to media campaigns from local healthcare organizations and found that care avoidance was lower among patients with good knowledge of the organization's safety protocols that were in place.

## Limitations

This study has some limitations that should be considered. The study, like previous studies focusing on avoidance, was cross-sectional. Therefore, it is not possible to establish a cause-

and-effect relationship. About 8% of our study sample reported they had avoided care when needed. Interestingly, this number was smaller than reports based on administrative data on visits to the emergency department and other acute care hospital-based care [16] as well as patient-reported avoidance. [7,24] One strength of the current design is that we capture patient-provided reasons for avoiding needed in-person healthcare regardless of venue. As discussed, most prior studies have focused uniquely on in-hospital care. Furthermore, it is a possibility that those who use ResearchMatch, our recruitment source, are less likely to avoid care than the general population, or that our survey sample does not capture the entire universe of those that might have needed acute in-person care. However, most previously published studies focused on administratively based data on visits to in-hospital care, including visits to emergency departments. In contrast, we attempted to capture the participants' avoidance of any type of in-person care, including outpatient and primary care. Since the latter represents a more representative picture of overall healthcare consumption, our results suggest that the impact of the pandemic on healthcare in-person visits might have been more limited. For example, in primary care, a large majority of in-person care visits were transferred to telemedicine visits [5,37]. While telehealth visits have been considered feasible alternatives for some patients [38], our study focused exclusively on avoidance of in-person care that patients deemed necessary.

Finally, although we reached a large sample, we used convenience sampling, and the study participants were largely female, White, and 55 years or older. Further, there may be some selection bias, as participants had to express interest in our study before receiving the link to the survey, and those who expressed interest may differ in some way from the broader population. Thus, care should be taken before generalizing these results to U.S. patients nationwide. However, care avoidance was significantly higher among younger and non-White individuals both in our study and in a study based on a U.S. nationally representative sample [7]. Our finding that care avoidance was higher among those with poor mental health was also in line with findings in another nationally representative sample [24]. It is also important to remember that our study took place before COVID-19 vaccine distribution began and state-specific lockdowns were lifted. However, we are not aware of any prior studies that have attempted to model risk factors to avoid in-patient care during the COVID-19 pandemic. Our best model explained only 11.8% of the overall variance, suggesting that additional, unmeasured variables may influence care avoidance. A possible source of bias was the exclusion of some health-related (need for emergency/urgent care, presence of chronic conditions, COVID-19 diagnosis) and some environmental (COVID-related news media) variables due to lack of a direct relationship with care avoidance and/or multicollinearity. Future studies exploring the possible role of these non-institutional factors may be warranted.

## Conclusions

Health care institutions can have a significant influence on patients' decisions to seek in-person care during the current pandemic. Our socio-ecological model of care avoidance seemed to be an effective tool in identifying key factors, suggesting that, while patient-related variables may be out of health care institutions' influence, malleable institution-related factors explained most of the care avoidance in our model. Thus, our study suggests that concrete steps, such as clear, effective communication of safety protocols and consistently focusing on enhancing patient healthcare experiences, can help to reduce care-avoiding behavior. In addition, patients' ability to afford care was significantly associated with care avoidance during the pandemic, further underscoring the urgent need to take steps to reduce health disparities and address social determinants of health both during and outside of public health emergencies.

Considering that many lost their jobs and their health insurance coverage during the pandemic, future public health efforts should focus on ensuring access to healthcare during public health emergencies. Stress caused by fear and anxiety was also an important determinant in our study and in prior pandemics. Future studies should examine the predictors of this fear and anxiety and how it can be mitigated.

## Author Contributions

**Conceptualization:** Bengt B. Arnetz, John vanSchagen, William Baer, Stacy Smith, Judith E. Arnetz.

**Data curation:** Bengt B. Arnetz, Courtney Goetz, Judith E. Arnetz.

**Formal analysis:** Bengt B. Arnetz, Courtney Goetz, Judith E. Arnetz.

**Funding acquisition:** Bengt B. Arnetz, John vanSchagen, William Baer, Judith E. Arnetz.

**Investigation:** Bengt B. Arnetz, Courtney Goetz, John vanSchagen, William Baer, Stacy Smith, Judith E. Arnetz.

**Methodology:** Bengt B. Arnetz, Courtney Goetz, John vanSchagen, William Baer, Stacy Smith, Judith E. Arnetz.

**Project administration:** Bengt B. Arnetz, Judith E. Arnetz.

**Resources:** Bengt B. Arnetz, Judith E. Arnetz.

**Software:** Bengt B. Arnetz, Courtney Goetz, Judith E. Arnetz.

**Supervision:** Bengt B. Arnetz, Judith E. Arnetz.

**Validation:** Bengt B. Arnetz, Courtney Goetz, William Baer, Stacy Smith, Judith E. Arnetz.

**Visualization:** Bengt B. Arnetz, Courtney Goetz, John vanSchagen, William Baer, Judith E. Arnetz.

**Writing – original draft:** Bengt B. Arnetz, Courtney Goetz, Judith E. Arnetz.

**Writing – review & editing:** Bengt B. Arnetz, Courtney Goetz, John vanSchagen, William Baer, Stacy Smith, Judith E. Arnetz.

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
