## [Decision Letter · Decision Letter 0]

8 Feb 2022

PONE-D-21-35375Patient reported factors associated with avoidance of in-person care during the COVID-19 pandemic: Results from a national survey.PLOS ONE

Dear Dr. Arnetz,

Thank you for submitting your manuscript to PLOS ONE. After careful consideration, we feel that it has merit but does not fully meet PLOS ONE’s publication criteria as it currently stands. Therefore, we invite you to submit a revised version of the manuscript that addresses the points raised during the review process.

Editor's comments:

1. Please revise the manuscript per the following reviewers' comments.

2. Please follow the PLoS One's guideline for preparing the manuscript.

3. Please cite (at least) five relevant PLoS Ones's articles.

We look forward to receiving your revised manuscript.

Kind regards,

Farzad Taghizadeh-Hesary

Academic Editor

PLOS ONE

Journal Requirements:

3. Please ensure that you include a title page within your main document. You should list all authors and all affiliations as per our author instructions and clearly indicate the corresponding author.

Reviewer #1 comments:

Overall an interesting piece of work that provides insights into patient perspective on in-person care avoidance during a pandemic. The results can potentially feed into care provider policies and improve services by addressing patient concerns and apprehensions during the current pandemic and beyond.

1. Methodology: It is not entirely clear whether telemedicine is being considered a sub-set of in-person care for the purpose of analysis in this study. Line # 337-338. For example, in primary care, a large majority of in-person care visits were transferred to telemedicine visits. Further, as in reference #12, (Alhadramy OM, Sr. The Structure and the Outcome of Telephone-Based Cardiac Consultations During Lockdown: A Lesson From COVID-19. Cureus. 2020), the conclusion states that: When standard face-to-face cardiac consultations are compromised, a structured telephone-based consultation is considered feasible, seems effective, and promising alternative method of delivering the utmost cardiac care to the community. It might be helpful if the authors mention that whether telemedicine was considered suitable alternative to physical in-person care, and how this has been dealt in the analysis.

2. Discussions:

a. It might be good to draw comparisons with results from other similar study at reference #7 (Czeisler ME, Marynak K, Clarke KEN, Salah Z, Shakya I, Thierry JM, et al. Delay or Avoidance of Medical Care Because of COVID-19-Related Concerns - United States, June 2020. MMWR Morb Mortal Wkly Rep. 2020). This would specifically be regarding

i. Is the current discussion on avoidance of in-person care, a subset of care avoidance discussed in reference #7?

ii. What is the possible influence of use of different tools for data collection on the results?

b. Line# 291-293: Our hypothesized model also included environment-related variables such as consuming COVID-related media news and the politics related to the pandemic, as we reasoned they might instill fear and panic, thereby influencing care avoidance. Although, as per the analysis, the variable did not have influence on patient decision in the analysis, this could have contributed to COVID related stress that had influence.

c. Kindly clarify if institutional factors like prioritization for COVID-19 patients and resource diversion could have been a factor for avoidance of in-person care?

d. Out of the seven key factors found to be associated with care avoidance, age, ability to afford care and HCO experience are not directly COVID-19 related. However, COVID-19 pandemic itself may be a confounding factor, exacerbating the inability to afford care. Kindly clarify if this has been explored.

e. As stated in the limitations, the sample recruitment is primarily from ResearchMatch which may not be representative of general population. As per the ResearchMatch site, persons of specific race are overrepresented, as are some of the States in the US. This could possibly skew the data either way. This may kindly be clarified if there is any published data from general population where care avoidance was stratified according to race and demography, and how the care avoidance seen in this study group compares with the general population data.

Others:

3. The proposed Figure 1 is an Onion diagram for groups of factors influencing care avoidance. However, the enlisted variables in the smaller circle are not a subset of those enlisted in the larger circle (sequentially). A Venn diagram or other similar options may offer better representation of the hypothesis used.

4. Line 13, please check if reference #11 is correct for the statement. It seems reference #12 might be more appropriate.

Reviewer #2 comments:

This cross-sectional online survey by Bengt B. Arnetz et al., 2021 examined reasons for the avoidance of in-person care among COVID-19 patients who reported that they needed it in the United States of America from late October to early December 2020.

The study used a 45-item online survey instrument administered via Qualtrics using the following variables:

Dependent (outcome) variable: care avoidance

Independent variables: categorized as patient related; environment related; and institutionally related variables

Out of the total of 3840 persons who anonymously responded to the survey, a reasonably large sample size (n=3373) fulfilled the inclusion criteria and were analyzed using multivariable logistic regression. Consent was obtained from participants and ethical clearance granted by the Michigan State University Institutional Review Board.

The authors made all data underlying the findings fully available. The data was tested for representativeness, analyzed using descriptive and inferential statistics which were rigorous and appropriate.

Results obtained revealed that 257 (7.6%) respondents avoided in-person care during the early phase of the pandemic citing reasons:

Patient related:

young age: inability to afford care; greater COVID-19 related stress

Environment related:

infrequent discussions about COVID with family and friends

Institution related:

negative patient health care experience rating; poor awareness of the institution’s safety protocol; low ratings of the institution’s effectiveness in communicating their safety protocol

Overall, the study predicted 11.9% of the variance in 30 care avoidance.

Discussions of the results were robust, citing similar studies conducted during previous SARS-CoV-1, H1N1 and MERS pandemics. However, the study, in my view, excluded important variables from the regression model: needed urgent care, needed emergency care, chronic conditions, COVID-19 diagnosis, general health rating, consuming COVID-related content on news media. Although the authors did this “due to lack of independent relationship with care avoidance”, nevertheless this exclusion may be an important source of bias to the study.

Conclusions are in line with the findings

Writing quality and clarity: Satisfactory

References: The manuscript employed the use of Harvard style referencing but requires editing to correct some errors noticed e.g., Listing of references: Shouldn’t this be in alphabetical order? Shouldn’t the journal name be italics? Shouldn’t the list of authors that are more than 5 be reflected as et al? e.g. reference number 7 could just be stated as Czeisler, M.E. et al. ( 2020).

I suggest the authors should revise Harvard referencing style and make necessary corrections.

Reviewer #3 comments:

The manuscript requires corrections throughout. The language needs improvement including sentence and paragraph structure in all areas of the manuscript. There is need for improvement to the referencing. Statements are made without references in several areas including the Discussion. The presentation of the results are very difficult to read and understand and requires significant improvement through revision. The Discussion is weak and the limitations are deficient.

Reviewers' comments:

Reviewer's Responses to Questions

**Comments to the Author**

1. Is the manuscript technically sound, and do the data support the conclusions?

Reviewer #1: Yes

Reviewer #2: Yes

Reviewer #3: Partly

2. Has the statistical analysis been performed appropriately and rigorously? 

Reviewer #1: Yes

Reviewer #2: Yes

Reviewer #3: I Don't Know

3. Have the authors made all data underlying the findings in their manuscript fully available?

Reviewer #1: Yes

Reviewer #2: Yes

Reviewer #3: Yes

4. Is the manuscript presented in an intelligible fashion and written in standard English?

Reviewer #1: Yes

Reviewer #2: Yes

Reviewer #3: No

5. Review Comments to the Author

Reviewer #1: Overall an interesting piece of work that provides insights into patient perspective on in-person care avoidance during a pandemic. The results can potentially feed into care provider policies and improve services by addressing patient concerns and apprehensions during the current pandemic and beyond.

1. Methodology: It is not entirely clear whether telemedicine is being considered a sub-set of in-person care for the purpose of analysis in this study. Line # 337-338. For example, in primary care, a large majority of in-person care visits were transferred to telemedicine visits. Further, as in reference #12, (Alhadramy OM, Sr. The Structure and the Outcome of Telephone-Based Cardiac Consultations During Lockdown: A Lesson From COVID-19. Cureus. 2020), the conclusion states that: When standard face-to-face cardiac consultations are compromised, a structured telephone-based consultation is considered feasible, seems effective, and promising alternative method of delivering the utmost cardiac care to the community. It might be helpful if the authors mention that whether telemedicine was considered suitable alternative to physical in-person care, and how this has been dealt in the analysis.

2. Discussions:

a. It might be good to draw comparisons with results from other similar study at reference #7 (Czeisler ME, Marynak K, Clarke KEN, Salah Z, Shakya I, Thierry JM, et al. Delay or Avoidance of Medical Care Because of COVID-19-Related Concerns - United States, June 2020. MMWR Morb Mortal Wkly Rep. 2020). This would specifically be regarding

i. Is the current discussion on avoidance of in-person care, a subset of care avoidance discussed in reference #7?

ii. What is the possible influence of use of different tools for data collection on the results?

b. Line# 291-293: Our hypothesized model also included environment-related variables such as consuming COVID-related media news and the politics related to the pandemic, as we reasoned they might instill fear and panic, thereby influencing care avoidance. Although, as per the analysis, the variable did not have influence on patient decision in the analysis, this could have contributed to COVID related stress that had influence.

c. Kindly clarify if institutional factors like prioritization for COVID-19 patients and resource diversion could have been a factor for avoidance of in-person care?

d. Out of the seven key factors found to be associated with care avoidance, age, ability to afford care and HCO experience are not directly COVID-19 related. However, COVID-19 pandemic itself may be a confounding factor, exacerbating the inability to afford care. Kindly clarify if this has been explored.

e. As stated in the limitations, the sample recruitment is primarily from ResearchMatch which may not be representative of general population. As per the ResearchMatch site, persons of specific race are overrepresented, as are some of the States in the US. This could possibly skew the data either way. This may kindly be clarified if there is any published data from general population where care avoidance was stratified according to race and demography, and how the care avoidance seen in this study group compares with the general population data.

Others:

3. The proposed Figure 1 is an Onion diagram for groups of factors influencing care avoidance. However, the enlisted variables in the smaller circle are not a subset of those enlisted in the larger circle (sequentially). A Venn diagram or other similar options may offer better representation of the hypothesis used.

4. Line 13, please check if reference #11 is correct for the statement. It seems reference #12 might be more appropriate

Reviewer #2: This cross-sectional online survey by Bengt B. Arnetz et al., 2021 examined reasons for the avoidance of in-person care among COVID-19 patients who reported that they needed it in the United States of America from late October to early December 2020.

The study used a 45-item online survey instrument administered via Qualtrics using the following variables:

Dependent (outcome) variable: care avoidance

Independent variables: categorized as patient related; environment related; and institutionally related variables

Out of the total of 3840 persons who anonymously responded to the survey, a reasonably large sample size (n=3373) fulfilled the inclusion criteria and were analyzed using multivariable logistic regression. Consent was obtained from participants and ethical clearance granted by the Michigan State University Institutional Review Board.

The authors made all data underlying the findings fully available. The data was tested for representativeness, analyzed using descriptive and inferential statistics which were rigorous and appropriate.

Results obtained revealed that 257 (7.6%) respondents avoided in-person care during the early phase of the pandemic citing reasons:

Patient related:

young age: inability to afford care; greater COVID-19 related stress

Environment related:

infrequent discussions about COVID with family and friends

Institution related:

negative patient health care experience rating; poor awareness of the institution’s safety protocol; low ratings of the institution’s effectiveness in communicating their safety protocol

Overall, the study predicted 11.9% of the variance in 30 care avoidance.

Discussions of the results were robust, citing similar studies conducted during previous SARS-CoV-1, H1N1 and MERS pandemics. However, the study, in my view, excluded important variables from the regression model: needed urgent care, needed emergency care, chronic conditions, COVID-19 diagnosis, general health rating, consuming COVID-related content on news media. Although the authors did this “due to lack of independent relationship with care avoidance”, nevertheless this exclusion may be an important source of bias to the study.

Conclusions are in line with the findings

Writing quality and clarity: Satisfactory

References: The manuscript employed the use of Harvard style referencing but requires editing to correct some errors noticed e.g., Listing of references: Shouldn’t this be in alphabetical order? Shouldn’t the journal name be italics? Shouldn’t the list of authors that are more than 5 be reflected as et al? e.g. reference number 7 could just be stated as Czeisler, M.E. et al. ( 2020).

I suggest the authors should revise Harvard referencing style and make necessary corrections.

Reviewer #3: The manuscript requires corrections throughout. The language needs improvement including sentence and paragraph structure in all areas of the manuscript. There is need for improvement to the referencing. Statements are made without references in several areas including the Discussion. The presentation of the results are very difficult to read and understand and requires significant improvement through revision. The Discussion is weak and the limitations are deficient.

6. PLOS authors have the option to publish the peer review history of their article (what does this mean?). If published, this will include your full peer review and any attached files.

Reviewer #1: **Yes: **Vineet Bhatia

Reviewer #2: **Yes: **Haruna Ismaila Adamu, MBBS; MPH; PhD

Reviewer #3: **Yes: **Glennis Andall-Brereton

---

## [Author Response · Author response to Decision Letter 0]

29 Mar 2022

Please see attachment Response to reviewer comments.

---

## [Decision Letter · Decision Letter 1]

1 Jun 2022

PONE-D-21-35375R1Patient reported factors associated with avoidance of in-person care during the COVID-19 pandemic: Results from a national survey.PLOS ONE

Dear Dr. Arnetz,

Thank you for submitting your manuscript to PLOS ONE. After careful consideration, we feel that it has merit but does not fully meet PLOS ONE’s publication criteria as it currently stands. Therefore, we invite you to submit a revised version of the manuscript that addresses the points raised during the review process. ===============================================================

ACADEMIC EDITOR: Please revise the manuscript per the reviewers' comments.===============================================================

We look forward to receiving your revised manuscript.

Kind regards,

Farzad Taghizadeh-Hesary

Academic Editor

PLOS ONE

Journal Requirements:

Additional Editor Comments:

None.

Reviewer #2:

This cross-sectional online survey was conducted in the United States by Arnetz, B. B., et al., 2022. from late October to early December 2020 with a questionnaire administered using ResearchMatch and Facebook. The study explored why patients avoided needed in-person care during the early phase of the COVID-19 pandemic in the United States from three perspectives: patient, external environment, and institution.

Dependent (outcome) variable: care avoidance

Independent variables: categorized as patient related; environment related; and institutionally related Variables

Out of the total of 3840 persons who anonymously responded to the survey, a reasonably large sample size (n=3373) fulfilled the inclusion criteria and were analyzed using multivariable logistic regression. Consent was obtained from participants and ethical clearance granted by the Michigan State University Institutional Review Board.

Statistical analysis was conducted using IBM SPSS Statistics version 26. A two-sided P value <0.05 represented statistical significance. A principal component analysis of the three safety rating variables gave a single factor solution with an alpha of .88, thus justifying the combination of the variables into an index

Results revealed that a total of 3840 patients completed the survey. Patients who reported that they had not needed care (n=423, 11.0%) and those not responding to the question (n=45, 1.2%) were excluded from this analysis, leaving a sample size of 3372. Out of a total of 3372 respondents who reported that they needed in-person care during the early phase of the pandemic, 257 (7.6%) avoided it. Patient-related variables associated with avoiding needed care included younger age (odds ratio (OR), 1.46, 95% CI 1.11 to 1.94, p<0.01; 53 <45 y/o vs 45+), inability to afford care (OR=1.65, 95% CI 1.17 to 2.34, p<0.01), and greater COVID-related stress (OR=1.36, CI 1.01 54 to 1.83, p<0.05). More frequent discussions about COVID with family and friends was the only significant environment-related avoidance of care variable (OR=1.39, 95% CI 1.01-1.91, p<.05). Institution-related care avoidance variables 57 included a negative patient healthcare experience rating (OR 1.83, 95% CI 1.38 to 2.42, 58 p<0.001), poor awareness of the institution’s safety protocol (OR=1.79, 95% CI 1.28 to 2.51, p<0.01), and low ratings of the institution’s effectiveness in communicating their safety protocol 60 (OR=3.45, 95% CI 1.94 to 6.12, p<0.001). The final model predicted 11.9% of the variance in care avoidance.

The authors made all data underlying the findings fully available. The data was tested for representativeness, analyzed using descriptive and inferential statistics which were rigorous and appropriate.

Discussions of the results were robust, citing similar studies conducted both within and outside Nepal.

Conclusions are in line with the findings

Writing quality and clarity: Satisfactory

Other observations:

1. Limitations of the study: The authors did well to mention the limitations of the study, including how these limitations should be addressed by future studies going forward

2. Inclusion/exclusion criteria clearly explained

References: The manuscript employed the use of Harvard style referencing but requires editing to correct some errors noticed e.g., Listing of references: Shouldn’t this be in alphabetical order? Shouldn’t the journal name be italics? Shouldn’t the list of authors that are more than 5 be reflected as et al?

I suggest the authors should revise the reference per the Vancouver referencing style and make necessary corrections.

Reviewer #4:

I thank the authors to address all the reviewer's comments. However, two minor comments are still remaining:

1. The reference style must be updated per the Vancouver style.

2. The following paper discuss on the Covid-19 impacts on the cancer care.

- https://www.ncbi.nlm.nih.gov/pmc/articles/PMC8184167/

It is suggested authors cite it and make a summarize this and similar articles regarding the Covid-19 impacts on patients cares in different disease entities (in 1-2 paragraphs in the Introduction).

Reviewers' comments:

Reviewer's Responses to Questions

**Comments to the Author**

1. If the authors have adequately addressed your comments raised in a previous round of review and you feel that this manuscript is now acceptable for publication, you may indicate that here to bypass the “Comments to the Author” section, enter your conflict of interest statement in the “Confidential to Editor” section, and submit your "Accept" recommendation.

Reviewer #1: All comments have been addressed

Reviewer #2: (No Response)

Reviewer #4: All comments have been addressed

2. Is the manuscript technically sound, and do the data support the conclusions?

Reviewer #1: Yes

Reviewer #2: Yes

Reviewer #4: Yes

3. Has the statistical analysis been performed appropriately and rigorously? 

Reviewer #1: Yes

Reviewer #2: Yes

Reviewer #4: Yes

4. Have the authors made all data underlying the findings in their manuscript fully available?

Reviewer #1: Yes

Reviewer #2: Yes

Reviewer #4: Yes

5. Is the manuscript presented in an intelligible fashion and written in standard English?

Reviewer #1: Yes

Reviewer #2: Yes

Reviewer #4: Yes

6. Review Comments to the Author

Reviewer #1: Thanks for addressing most of the comments and providing justification response to others. Best wishes!

Reviewer #2: This cross-sectional online survey was conducted in the United States by Arnetz, B. B., et al., 2022. from late October to early December 2020 with a questionnaire administered using ResearchMatch and Facebook. The study explored why patients avoided needed in-person care during the early phase of the COVID-19 pandemic in the United States from three perspectives: patient, external environment, and institution.

.

Dependent (outcome) variable: care avoidance

Independent variables: categorized as patient related; environment related; and institutionally related Variables

Out of the total of 3840 persons who anonymously responded to the survey, a reasonably large sample size (n=3373) fulfilled the inclusion criteria and were analyzed using multivariable logistic regression. Consent was obtained from participants and ethical clearance granted by the Michigan State University Institutional Review Board.

Statistical analysis was conducted using IBM SPSS Statistics version 26. A two-sided P value <0.05 represented statistical significance. A principal component analysis of the three safety rating variables gave a single factor solution with an alpha of .88, thus justifying the combination of the variables into an index

Results revealed that a total of 3840 patients completed the survey. Patients who reported that they had not needed care (n=423, 11.0%) and those not responding to the question (n=45, 1.2%) were excluded from this analysis, leaving a sample size of 3372. Out of a total of 3372 respondents who reported that they needed in-person care during the early phase of the pandemic, 257 (7.6%) avoided it. Patient-related variables associated with avoiding needed care included younger age (odds ratio (OR), 1.46, 95% CI 1.11 to 1.94, p<0.01; 53 <45 y/o vs 45+), inability to afford care (OR=1.65, 95% CI 1.17 to 2.34, p<0.01), and greater COVID-related stress (OR=1.36, CI 1.01 54 to 1.83, p<0.05). More frequent discussions about COVID with family and friends was the only significant environment-related avoidance of care variable (OR=1.39, 95% CI 1.01-1.91, p<.05). Institution-related care avoidance variables

57 included a negative patient healthcare experience rating (OR 1.83, 95% CI 1.38 to 2.42,

58 p<0.001), poor awareness of the institution’s safety protocol (OR=1.79, 95% CI 1.28 to 2.51, p<0.01), and low ratings of the institution’s effectiveness in communicating their safety protocol 60 (OR=3.45, 95% CI 1.94 to 6.12, p<0.001). The final model predicted 11.9% of the variance in care avoidance.

The authors made all data underlying the findings fully available. The data was tested for representativeness, analyzed using descriptive and inferential statistics which were rigorous and appropriate.

Discussions of the results were robust, citing similar studies conducted both within and outside Nepal.

Conclusions are in line with the findings

Writing quality and clarity: Satisfactory

Other observations:

1. Limitations of the study: The authors did well to mention the limitations of the study, including how these limitations should be addressed by future studies going forward

2. Inclusion/exclusion criteria clearly explained

References: The manuscript employed the use of Harvard style referencing but requires editing to correct some errors noticed e.g., Listing of references: Shouldn’t this be in alphabetical order? Shouldn’t the journal name be italics? Shouldn’t the list of authors that are more than 5 be reflected as et al?

I suggest the authors should revise Harvard referencing style and make necessary corrections.

Reviewer #4: I thank the authors to address all the reviewer's comments. However, two minor comments are still remaining:

1. The reference style must be updated per the Vancouver style.

2. The following paper discuss on the Covid-19 impacts on the cancer care.

- https://www.ncbi.nlm.nih.gov/pmc/articles/PMC8184167/

It is suggested authors cite it and make a summarize this and similar articles regarding the Covid-19 impacts on patients cares in different disease entities (in 1-2 paragraphs in the Introduction).

7. PLOS authors have the option to publish the peer review history of their article (what does this mean?). If published, this will include your full peer review and any attached files.

Reviewer #1: **Yes: **Vineet Bhatia

Reviewer #2: **Yes: **Haruna Ismaila ADAMU, MD; MPH; PhD; MACE

Reviewer #4: No

---

## [Author Response · Author response to Decision Letter 1]

22 Jun 2022

Manuscript PONE-D-21-35375R1: Response to reviewer comments

Dear Editor:

Our responses to the reviewer comments are in italics below each comment. Our thanks to the reviewers for their comments that have helped us to improve our paper.

Reviewer #1: Thanks for addressing most of the comments and providing justification response to others. Best wishes!

Response: Thank you!

Reviewer #2: This cross-sectional online survey was conducted in the United States by Arnetz, B. B., et al., 2022. from late October to early December 2020 with a questionnaire administered using ResearchMatch and Facebook. The study explored why patients avoided needed in-person care during the early phase of the COVID-19 pandemic in the United States from three perspectives: patient, external environment, and institution.

Dependent (outcome) variable: care avoidance

Independent variables: categorized as patient related; environment related; and institutionally related Variables

Out of the total of 3840 persons who anonymously responded to the survey, a reasonably large sample size (n=3373) fulfilled the inclusion criteria and were analyzed using multivariable logistic regression. Consent was obtained from participants and ethical clearance granted by the Michigan State University Institutional Review Board.

Statistical analysis was conducted using IBM SPSS Statistics version 26. A two-sided P value <0.05 represented statistical significance. A principal component analysis of the three safety rating variables gave a single factor solution with an alpha of .88, thus justifying the combination of the variables into an index

Results revealed that a total of 3840 patients completed the survey. Patients who reported that they had not needed care (n=423, 11.0%) and those not responding to the question (n=45, 1.2%) were excluded from this analysis, leaving a sample size of 3372. Out of a total of 3372 respondents who reported that they needed in-person care during the early phase of the pandemic, 257 (7.6%) avoided it. Patient-related variables associated with avoiding needed care included younger age (odds ratio (OR), 1.46, 95% CI 1.11 to 1.94, p<0.01; 53 <45 y/o vs 45+), inability to afford care (OR=1.65, 95% CI 1.17 to 2.34, p<0.01), and greater COVID-related stress (OR=1.36, CI 1.01 54 to 1.83, p<0.05). More frequent discussions about COVID with family and friends was the only significant environment-related avoidance of care variable (OR=1.39, 95% CI 1.01-1.91, p<.05). Institution-related care avoidance variables 57 included a negative patient healthcare experience rating (OR 1.83, 95% CI 1.38 to 2.42, 58 p<0.001), poor awareness of the institution’s safety protocol (OR=1.79, 95% CI 1.28 to 2.51, p<0.01), and low ratings of the institution’s effectiveness in communicating their safety protocol 60 (OR=3.45, 95% CI 1.94 to 6.12, p<0.001). The final model predicted 11.9% of the variance in care avoidance.

The authors made all data underlying the findings fully available. The data was tested for representativeness, analyzed using descriptive and inferential statistics which were rigorous and appropriate.

Discussions of the results were robust, citing similar studies conducted both within and outside Nepal.

Conclusions are in line with the findings

Writing quality and clarity: Satisfactory

Other observations:

1. Limitations of the study: The authors did well to mention the limitations of the study, including how these limitations should be addressed by future studies going forward

2. Inclusion/exclusion criteria clearly explained

References: The manuscript employed the use of Harvard style referencing but requires editing to correct some errors noticed e.g., Listing of references: Shouldn’t this be in alphabetical order? Shouldn’t the journal name be italics? Shouldn’t the list of authors that are more than 5 be reflected as et al?

I suggest the authors should revise the reference per the Vancouver referencing style and make necessary corrections.

Response: Per the journal submission guidelines (https://journals.plos.org/plosone/s/submission-guidelines), PLOS ONE requires that references follow the Vancouver style, which we have utilized in our paper. However, based on comments by reviewer #4, our entire reference list has been updated to the most recent Vancouver style. According to guidelines, references are in the order in which they are cited, not in alphabetical order. Journal names are based on the NCBI database abbreviations and are not italicized.

Reviewer #4:

I thank the authors to address all the reviewer's comments. However, two minor comments are still remaining:

1. The reference style must be updated per the Vancouver style.

Response: Please see our response to the previous point (Reviewer #2). 

2. The following paper discuss on the Covid-19 impacts on the cancer care.

- https://www.ncbi.nlm.nih.gov/pmc/articles/PMC8184167/

It is suggested authors cite it and make a summarize this and similar articles regarding the Covid-19 impacts on patients cares in different disease entities (in 1-2 paragraphs in the Introduction).

Response: Our thanks to the reviewer for this interesting paper and for this suggestion. We have added a paragraph in the Introduction, lines 40-54, where we review studies of the impact of the pandemic on specific disease conditions. 

Response: We have revised our Figure using the PACE digital diagnostic tool.

---

## [Decision Letter · Decision Letter 2]

25 Jul 2022

Patient reported factors associated with avoidance of in-person care during the COVID-19 pandemic: Results from a national survey.

PONE-D-21-35375R2

Dear Dr. Arnetz,

We’re pleased to inform you that your manuscript has been judged scientifically suitable for publication and will be formally accepted for publication once it meets all outstanding technical requirements.

Kind regards,

Farzad Taghizadeh-Hesary

Academic Editor

PLOS ONE

Reviewers' comments:

Reviewer's Responses to Questions

**Comments to the Author**

1. If the authors have adequately addressed your comments raised in a previous round of review and you feel that this manuscript is now acceptable for publication, you may indicate that here to bypass the “Comments to the Author” section, enter your conflict of interest statement in the “Confidential to Editor” section, and submit your "Accept" recommendation.

Reviewer #2: All comments have been addressed

Reviewer #4: All comments have been addressed

2. Is the manuscript technically sound, and do the data support the conclusions?

Reviewer #2: Yes

Reviewer #4: Yes

3. Has the statistical analysis been performed appropriately and rigorously? 

Reviewer #2: Yes

Reviewer #4: Yes

4. Have the authors made all data underlying the findings in their manuscript fully available?

Reviewer #2: Yes

Reviewer #4: Yes

5. Is the manuscript presented in an intelligible fashion and written in standard English?

Reviewer #2: Yes

Reviewer #4: Yes

6. Review Comments to the Author

Reviewer #2: (No Response)

Reviewer #4: (No Response)

7. PLOS authors have the option to publish the peer review history of their article (what does this mean?). If published, this will include your full peer review and any attached files.

Reviewer #2: **Yes: **Haruna Ismaila ADAMU, MD; MPH; PhD; MACE

Reviewer #4: No

---

## [Editor Report · Acceptance letter]

28 Jul 2022

PONE-D-21-35375R2 

Patient-reported factors associated with avoidance of in-person care during the COVID-19 pandemic: Results from a national survey 

Dear Dr. Arnetz:

I'm pleased to inform you that your manuscript has been deemed suitable for publication in PLOS ONE. Congratulations! Your manuscript is now with our production department. 

Kind regards, 

on behalf of

Dr. Farzad Taghizadeh-Hesary 

Academic Editor

PLOS ONE